# Transcriptomic Analysis of Shiga Toxin-Producing *Escherichia coli* during Initial Contact with Cattle Colonic Explants

**DOI:** 10.3390/microorganisms8111662

**Published:** 2020-10-27

**Authors:** Zachary R. Stromberg, Rick E. Masonbrink, Melha Mellata

**Affiliations:** 1Department of Food Science and Human Nutrition, Iowa State University, Ames, IA 50011, USA; zstrom@iastate.edu; 2Genome Informatics Facility, Iowa State University, Ames, IA 50011, USA; remkv6@iastate.edu

**Keywords:** adherence, cattle, colon, *Escherichia coli*, flagellin, STEC, transcriptome

## Abstract

Foodborne pathogens are a public health threat globally. Shiga toxin-producing *Escherichia coli* (STEC), particularly O26, O111, and O157 STEC, are often associated with foodborne illness in humans. To create effective preharvest interventions, it is critical to understand which factors STEC strains use to colonize the gastrointestinal tract of cattle, which serves as the reservoir for these pathogens. Several colonization factors are known, but little is understood about initial STEC colonization factors. Our objective was to identify these factors via contrasting gene expression between nonpathogenic *E*. *coli* and STEC. Colonic explants were inoculated with nonpathogenic *E*. *coli* strain MG1655 or STEC strains (O26, O111, or O157), bacterial colonization levels were determined, and RNA was isolated and sequenced. STEC strains adhered to colonic explants at numerically but not significantly higher levels compared to MG1655. After incubation with colonic explants, flagellin (*fliC*) was upregulated (log_2_ fold-change = 4.0, *p* < 0.0001) in O157 STEC, and collectively, Lon protease (*lon*) was upregulated (log_2_ fold-change = 3.6, *p* = 0.0009) in STEC strains compared to MG1655. These results demonstrate that H7 flagellum and Lon protease may play roles in early colonization and could be potential targets to reduce colonization in cattle.

## 1. Introduction

Shiga toxin-producing *Escherichia coli* (STEC) strains are gastrointestinal pathogens that cause mild to severe bloody diarrhea in humans [1]. Shiga toxin is the main virulence factor in STEC, encoded by the *stx* gene, and exists in two major types as *stx*_1_ and *stx*_2_ [2]. Clinical disease caused by STEC infections can occur in humans at any age, but children less than five years old are particularly susceptible [3]. Approximately 15% of STEC infections will progress to the severe disease, hemolytic uremic syndrome [4]. Although not all STEC strains cause severe disease, some STEC serogroups are predominantly found in patients with O26, O111, and O157 STEC, collectively accounting for an estimated 86–89% of STEC illnesses in the U.S. [5,6]. Common sources of human infection include contaminated foods such as beef, dairy, and produce [7]. The reduction of these pathogens in animals, food processing plants, and elimination from food products is important for protecting public health [8].

Ruminants, specifically cattle, constitute a significant reservoir of STEC and often carry this human pathogen asymptomatically, although a few clinical cases in cattle have been reported [9,10]. STEC strains use several factors to survive and colonize the gastrointestinal tract [11]. The adherence factor intimin (*eae*), and the type III secretion of the translocated intimin receptor and other effectors, play a major role in colonizing the bovine ileum and colon and causing attaching/effacing lesions [12,13,14]. Other colonization factors, such as flagella [15] and fimbria [16], have been identified as well. Some cattle colonized with STEC shed this bacterium in feces, but shedding frequencies and concentrations can greatly vary across herds [17]. Cattle shedding O157 STEC at high levels (≥10^4^ CFU/g of feces) can considerably spread this pathogen to other animals in the feedlot and subsequently contaminate carcasses and beef [18,19]. Recently, some non-O157 STEC serogroups have also been shown to be shed at high levels in fecal samples [20,21]. Thus, reducing colonization in cattle could reduce the spread between animals and contamination during processing.

Though it is well known that intimin and the translocated intimin receptor are involved in adherence and attaching/effacing lesion formation [22], our knowledge of initial colonization factors is limited. Therefore, understanding which factors are essential for initial colonization of the bovine intestine will facilitate improved interventions. The overall goals of this study were to compare the abilities of O26, O111, and O157 STEC to colonize cattle colonic explants and use bacterial RNA-seq to determine which genes are differentially regulated in STEC compared with *E*. *coli* K-12 during colonization of cattle colonic explants. Detection of conserved genes found in all strains that are differentially regulated in STEC compared to *E*. *coli* K-12 could lead to improved interventions that reduce STEC colonization of the gastrointestinal tract while maintaining resident *E*. *coli* strains of the microbiota.

## 2. Materials and Methods

### 2.1. Bacterial Strains

The STEC strains DEC10E (O26:H11, *stx*_1_, *eae*) isolated from a calf [23], DEC8B (O111:H8, *stx*_1_, stx_2_, *eae*) isolated from a human patient [23], and EDL933 (O157:H7, *stx*_1_, *stx*_2_, *eae*) isolated from ground beef [24] were obtained from Dr. Shannon Manning (Michigan State University STEC Center, Lansing, MI, USA). The *E*. *coli* K-12 strain MG1655 (OR:H48) was from our laboratory collection.

### 2.2. Cattle Colonic Explants

Bovine colonic explants and bacterial strains were prepared using methods previously described [13] with minor modifications. In 2017, tissues were obtained from three 18-month-old steers of approximately 1200 lb on three separate days (June 14, June 30, and July 24) at a meat processor with each animal representing a single experiment. A 0.3 m section of the centripetal coil was taken near the junction of the centrifugal and centripetal coils of the spiral colon following Iowa State University and USDA guidelines for collecting tissues from postmortem animals. The time-lapse between death and initiation of explant cultures was approximately 30 min. Collected tissues were washed with cold saline, immersed in CO_2_-independent medium (Gibco, Waltham, MA, USA) with gentamicin (50 µg/mL), penicillin (100 U/mL), and streptomycin (100 µg/mL), and transported to the laboratory on ice. Tissues were removed from media, thoroughly washed in saline, cut into 4 mm^2^ explants, and placed mucosal side up onto biopsy foam pads (4 explants per pad). The foam pads containing explants were placed one per well in six-well polystyrene tissue culture plates containing RPMI 1640 complete medium (RPMI 1640 supplemented with 10% fetal bovine serum, 0.25% lactalbumin hydrolysate, 0.10 μg/mL human insulin, and 1% D- + -mannose) (Figure 1). For the inoculation of explants, bacterial strains were grown statically overnight in trypticase soy broth (TSB) at 37 °C and then transferred 1:10 into fresh TSB for 6 h. Strains were washed and resuspended in RPMI 1640 complete medium and explants were inoculated with 2 × 10^7^ CFU in accordance with protocols approved by Iowa State University (protocol number 16-D/I-0015-A/H). The inoculum concentration was confirmed by serial dilution and plating on MacConkey agar. After inoculation, six-well plates were incubated for 2 h at 37 °C in 5% CO_2_ on a rocker (16 cycles/min).

### 2.3. Bacterial Adherence to Explants

Bacterial adherence to explants was determined using one explant per bacterial strain for each experiment (3 per strain) after the 2 h incubation. The explant was first weighed and then placed in a 2 oz. Whirl-Pak bag and homogenized in PBS. Samples were serially diluted in PBS, plated on MacConkey agar, and incubated at 37 °C for 18 h. To quantify the *E*. *coli* associated per explant, CFUs were recorded and divided by the weight of the explant (CFU/g).

### 2.4. RNA Extraction, Library Generation, and Sequencing

After the 2 h incubation, the remaining explants were snap-frozen in liquid nitrogen and stored at –80 °C until needed. For RNA extraction, samples were lysed and homogenized in 2 mL lysing matrix bead-beating tubes (MP Biomedicals, Santa Ana, CA, USA). The remaining extraction was performed using a PureLink RNA Mini Kit (Life Technologies, Carlsbad, CA, USA) with on-column DNase treatment. RNA quantity was determined using a Qubit fluorometer (Thermo Fisher, Waltham, MA, USA). RNA quality was determined using a 2100 Bioanalyzer (Agilent, Santa Clara, CA, USA) and samples with an RNA integrity number greater than 7 were used for mRNA-seq. Bacterial rRNA was depleted using Ribo-Zero (Illumina, San Diego, CA, USA) and samples were sequenced on a HiSeq 3000 (Illumina, San Diego, CA, USA) generating 150 bp single-end reads. RNA-seq data were deposited in the NCBI Sequence Read Archive (SRA) under accession number PRJNA667523.

### 2.5. RNA-seq Analysis and Detection of Differential Gene Expression

Genomes, genomic gff’s, and protein fastas were downloaded from NCBI for *E*. *coli* strains (accessions): DEC10E (GCF_000250075.1), DEC8B (GCF_000249815.1), MG1655 (GCF_000005845.2), and EDL933 (GCF_000732965.1) [25,26]. Raw reads were trimmed for adapters and quality using trim_galore version 0.4.5 (Babraham Institute, Babraham, UK). Reads were mapped to their respective *E*. *coli* genome using Hisat2 version 2.1.0. Read counts were obtained using featureCounts from the Subread package version 1.6.0 (WEHI, Melbourne, Australia). Orthologous gene families were identified using Orthofinder version 2.2.0 [27]. Single-copy orthologous genes were subjected to a differential gene expression analysis using DESeq2 (1.20.0) with a cutoff of adjusted *p* < 0.05 [28]. In the differential gene expression analysis, STEC strains were compared to MG1655 both as individual strains and as a collective group.

### 2.6. Statistical Analysis

Statistical analyses were performed using GraphPad Prism version 7 (San Diego, CA, USA). An ANOVA followed by Tukey’s method for multiple means comparison was used to compare *E*. *coli* strains in adherence assays. *p* < 0.05 was considered significant.

## 3. Results and Discussion

### 3.1. STEC Strains and MG1655 Adhered to Colonic Explants

STEC adherence to the colonic epithelium is a crucial step for colonizing cattle [29]. The proposed timeline of STEC adherence begins with initial interaction with the extracellular membrane [30], which was the focus of our study. To quantify this early interaction, a total of three bovine colonic explants were inoculated with *E*. *coli* strains and bacterial adherence was evaluated after 2 h. Previous studies demonstrated that 2 h is sufficient to produce initial adherence factors such as fimbriae and pili in *E*. *coli* and to allow for attachment to cell cultures [30,31]. Uninoculated explants were used as a control to ensure that no background *E*. *coli* or other *Enterobacteriaceae* organisms remained on the tissues. No growth was observed on MacConkey agar from plating the uninoculated colonic explant samples. The mean adherence level of the STEC strains ranged from 6.5 to 7.2 log CFU/g, while for MG1655 it was 4.2 log CFU/g (Figure 2). Although STEC strains adhered at numerically higher levels, no significant differences were detected between the strains. Variation in adherence levels between experiments was observed, likely explained by the variability in host tissues collected from three different steers. Adherence assays that use cell lines have less variation [32]; however, there is no commercial source of cattle colonic epithelial cells. Thus, researchers have relied upon explant cultures to assess STEC colonization in the laboratory [13,33,34,35]. Development of stable, immortalized intestinal epithelial cell lines such as that recently reported from the ileum of a calf [36] are needed and would help standardize results for several gastrointestinal pathogens in cattle.

In cattle, STEC organisms can be found throughout the gastrointestinal tract [37]. The bovine internal temperature is 39 °C; however, explants were cultured and infected at 37 °C based on established conditions in a previous study [13], which may be a limitation of our study in mimicking physiological conditions. In a previous study, calf colonic explants were inoculated with 10^7^ CFU of O157 STEC strains, incubated for 3 h, and displayed similar adherence levels compared with our current study [35]. Most studies have only explored O157 STEC adherence to colonic explants, while we tested both O157 and non-O157 STEC strains. In one of the few studies that did test both O157 and non-O157 STEC strains, it was demonstrated that O26, O111, and O157 adhered to the terminal ileum, colon, and rectum from calves after an 8 h incubation and formed attaching/effacing lesions [34]. Demonstration of STEC adherence to bovine tissues is critical for understanding tissue tropism and adherence factors, but could also be used as a model system to evaluate interventions. Previously, the treatment of colonic explants with TNF-α reduced adhesion of O157 STEC [38]. An intervention was not evaluated in the current study; however, future assessments could evaluate the use of interventions to inhibit O157 and non-O157 STEC attachment in this model system.

### 3.2. H7 Flagellin (fliC) and Lon Protease (lon) Upregulated during Incubation with Colonic Explants

We sought to identify differentially regulated genes between STEC strains and the *E*. *coli* K-12 strain MG1655 using RNA-seq reads. We first compared gene expression from individual STEC strains to MG1655 (i.e., DEC10E vs. MG1655, DEC8B vs. MG1655, and EDL933 vs. MG1655). Single-copy orthogroups were identified among strain comparisons with 3452 orthogroups found in DEC10E vs. MG1655, 3473 orthogroups in DEC8B vs. MG1655, and 3460 orthogroups in EDL933 vs. MG1655 (Appendix A). No significant changes in gene expression were observed in DEC10E nor DEC8B when compared with MG1655. This suggests that the genes shared between these strains responded in a similar manner during the 2 h incubation with bovine explants. For strain EDL933, flagellin (*fliC*) was significantly upregulated (log_2_ fold-change = 4.0, adjusted *p* < 0.0001) in O157 STEC compared to MG1655 (Appendix A). The RNA-seq count data was consistent between the three explants used with EDL933 having higher counts compared to MG1655 (Figure 3A). Flagellin is the main component of the bacterial flagellum and forms the filament [39]. These results support an earlier finding that H7 flagellin is expressed during contact with cattle intestinal epithelial cells [15]. In addition, the deletion of *fliC*_H7_ in EDL933 was previously shown to reduce adherence to bovine explants [40]. In vivo, cattle inoculated with an O157 STEC strain lacking flagellar expression had reduced levels of fecal shedding compared to the wild-type strain [41]. The induced production of flagellin-specific IgG and IgA antibodies and reducing colonization rates was previously demonstrated in cattle immunized intramuscularly with H7 flagellum [42]. Overall, these findings demonstrate that H7 flagellin functions as an adhesin in the gastrointestinal tract of cattle and a potential vaccine candidate to reduce STEC colonization. In addition to results found in cattle, *fliC*_H7_ in EDL933 has also been shown to play an important role in initial attachment to spinach leaves and glass [43], suggesting that H7 flagellum is a critical adherence factor for both the attachment to biotic and abiotic material. Other flagellin H-types such as *fliC*_H12_ have been shown to facilitate attachment to glass and stainless steel surfaces [44], but involvement in colonization of cattle is unclear. In the current study, we demonstrated that only *fliC*_H7_ in EDL933 was significantly upregulated, while no significant differences were found with *fliC*_H11_ in DEC10E and *fliC*_H8_ in DEC8B when compared to *fliC*_H48_ in MG1655. Thus, H7 flagellin appears to be a strong intervention candidate to reduce O157 STEC colonization, though few candidates exist that can reduce non-O157 STEC colonization in cattle.

Gene expression from STEC strains collectively was compared to MG1655 (i.e., STEC vs. MG1655). There were 3198 single-copy orthogroups identified among the STEC vs. MG1655 comparison. In this comparison, flagellin (*fliC*), ferrous iron transport protein A (*feoA*), ketol-acid reductoisomerase (*ilvC*), and nucleoid occlusion protein (*slmA*) were > 5-fold upregulated with a *p* < 0.05, but had an adjusted *p* > 0.05 likely due to variability between explants and between STEC strains. Interestingly, flagellin (Figure 3A) was identified in the EDL933 vs MG1655 comparison, but the relatively low reads from the other STEC strains DEC10E and DEC8B lend evidence that not all flagellin types are initial adherence factors. The ferrous iron transport protein A (*feoA*, Figure 3B) has a potential role in iron transport, which is an important nutrient for microorganisms [45]. However, relatively low levels of reads for DEC10E may have contributed to the adjusted *p* > 0.05. A relatively high level of reads were identified in some STEC, especially DEC8B, for the oxidoreductase enzyme ketol-acid reductoisomerase (*ilvC*, Figure 3C). However, in the individual comparison of DEC8B vs MG1655, the adjusted *p* = 0.051 and did not pass our cutoff. In addition, the normalized RNA-seq reads for nucleoid occlusion factor (*slmA*), which is involved in regulating cell division [46], was 6.2-fold higher in STEC compared to MG1655 (Figure 3D), but was not significant (adjusted *p* > 0.05).

Collectively, Lon protease (*lon*) was significantly upregulated (log_2_ fold-change = 3.6, adjusted *p* = 0.0009) in STEC strains compared to MG1655 (Appendix A). However, there was variation between the explant samples (Figure 3E). For the RNA-seq counts, DEC10E remained relatively low across all three explants, DEC8B and EDL933 had higher levels than MG1655 for Explants 1 and 3, while numerically higher counts were observed for MG1655 from explant 2 (Figure 3A). As with the adherence levels, variation in this data may be due to the variability in the tissue collected from three different steers. The role of *lon* in colonization remains unclear; future studies are needed to compare colonization with wild-type and *lon* knockout mutants to determine whether *lon* is involved in colonization. In general, Lon protease is a highly conserved ATP-dependent protease responsible for the degradation of abnormal and misfolded proteins [47]. In *E*. *coli*, there are several regulatory functions of Lon protease that are related to stress response, including roles in amino acid starvation [48], defense against reactive oxygen species [49], and heat shock [50,51]. Thus, *lon* may have been upregulated in STEC strains to help survive stress encountered after the inoculation of bovine explants. Interestingly in STEC, Lon protease degrades the regulatory protein Ler, which activates genes in the locus of enterocyte effacement [52,53], which suggests that it contributes to regulation and timing of attaching/effacing lesions.

Unlike many studies that have evaluated gene expression changes under different conditions, the current study evaluated differences between strains. This approach was used to explore whether conserved genes found in all strains were differentially regulated in STEC compared to *E*. *coli* K-12. This approach may account for why we found only a few differentially regulated genes. In addition, the use of the complex RPMI 1640 medium during incubation may have contributed to the homogenous response observed by RNA-seq. The evaluation of only one incubation time may have resulted in similar responses between strains. To our knowledge, no study has systemically evaluated a time-series experiment to determine STEC adherence and gene expression over time during contact with bovine explants, which warrants future investigation. Other studies have used RNA-seq to explore the host response to STEC infection. Bando et al. found that a STEC strain from a human patient induced a distinct response in the Caco-2 colonic cell line compared with a strain isolated from bovine feces [54]. In another study, 351 differential regulated genes were identified throughout the bovine intestine of cattle shedding high levels of O157 STEC compared to cattle negative for O157 STEC [55]. In future studies, dual RNA-seq of the host and STEC could be used as a powerful approach to simultaneously understand both host and bacterial changes in gene expression, evaluate host–STEC interactions, and potentially combat STEC colonization of the bovine gastrointestinal tract.

## 4. Conclusions

In this study, cattle colonic explants were inoculated with STEC and *E*. *coli* K-12 strains and evaluated for colonization and bacterial gene expression. STEC strains adhered at numerically but not significantly higher levels compared to MG1655. Flagellin (*fliC*) was upregulated (log_2_ fold-change = 4.0, adjusted *p* < 0.0001) in O157 STEC compared to MG1655. Collectively, Lon protease (*lon*) was upregulated (log_2_ fold-change = 3.6, adjusted *p* = 0.0009) in STEC strains compared to MG1655. These results demonstrate that H7 flagellum and Lon protease potentially play roles in early colonization and could be intervention targets to reduce colonization in cattle.

## Figures and Tables

**Figure 1 microorganisms-08-01662-f001:**
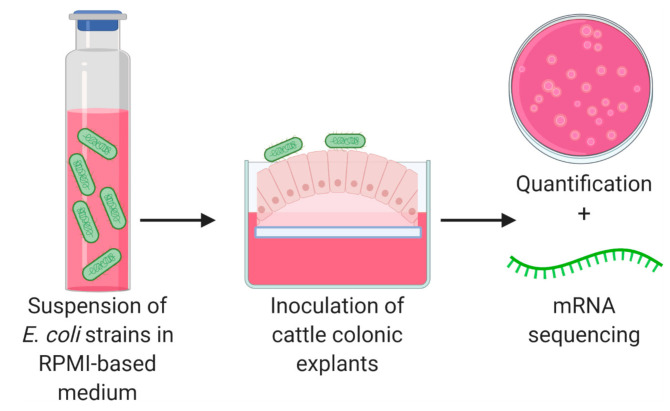
Experimental overview of bovine colonic explants inoculated with *Escherichia coli*. After a 2 h incubation, explants were either homogenized for bacterial enumeration on MacConkey agar or frozen for RNA extraction and sequencing.

**Figure 2 microorganisms-08-01662-f002:**
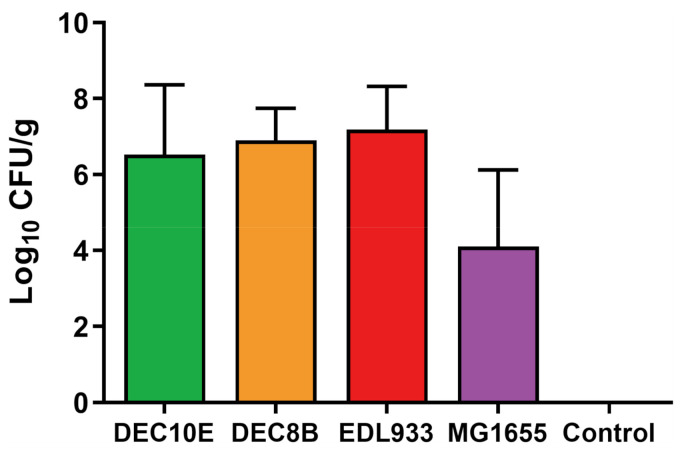
Bacterial adherence to cattle colonic explants. Bacterial counts of Shiga toxin-producing *Escherichia coli* (STEC) strains DEC10E (green), DEC8B (orange), and EDL933 (red), *E*. *coli* K-12 MG1655 (purple), and a control (uninoculated) were obtained after 2 h of incubation with explant cultures. Bars represent the mean of three individual experiments, and error bars represent standard deviation.

**Figure 3 microorganisms-08-01662-f003:**
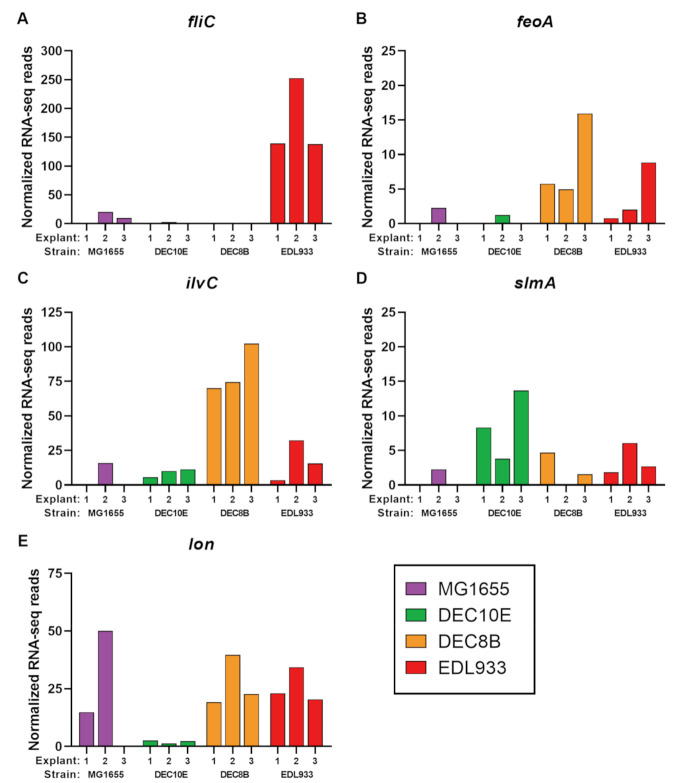
Normalized RNA-seq reads. (**A**) Significantly upregulated flagellin gene (*fliC*) in the EDL933 vs. MG1655 comparison, (**B**) ferrous iron transport protein A gene (*feoA*), (**C**) ketol-acid reductoisomerase gene (*ilvC*), (**D**) nucleoid occlusion gene (*slmA*), and (**E**) significantly upregulated Lon protease gene (*lon*) in the Shiga toxin-producing *Escherichia coli* (STEC) vs. MG1655 comparison. Three biological replicates (Explants 1, 2, and 3) were inoculated with STEC strains DEC10E (green), DEC8B (orange), and EDL933 (red), and *E*. *coli* K-12 strain MG1655 (purple).

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
