# Peer review of "Transcriptomic Analysis of Shiga Toxin-Producing Escherichia coli during Initial Contact with Cattle Colonic Explants"

_microorganisms, 2020, doi:10.3390/microorganisms8111662_

Round 1

Reviewer 1 Report

Dear all

First of all congratulations to the authors for this excellent piece of scientific work. Identifying the factors that allow STEC (and other EHEC) bacteria to colonize the gastrointestinal tract of cattle is a hot topic that interests not only specialized audience but also a range of scientists and other professionals related to healthcare epidemiology and infection control, animal husbandry, food industry etc

I only have a few minor comments/suggestions that I hope the authors will find useful:

L45: to an unfamiliar reader this sounds like all cattle carrying EHEC shed the bacterium, however this is not the case as not all cattle carrying EHEC shed it. Maybe make it a bit more clear?

L50-51: add a reference to this statement

L66: maybe add a few details about the different days the tissues were obtained from the steers

L87: I am a bit curious on why you incubated for 2h (and not more or less). This is probably something I do not know of but 1-2 more sentences would help non-specialists reading the article. Is it 2h because 1h is not enough and more than 2h is not necessary?

L104: raw? Reads were trimmed

L138-139: related to my previous comment: is the fact that 3h incubation shows similar results suggesting that 2h is enough for the initial STEC colonization? (question/just thinking out loud here): are there any studies that used a time series experiment to show how the adherence progresses

L169-173: personally, I find the biotic/abiotic comparison a bit far fetched – but please do not remove it on my account; I simply say that I would be more interested in a comparison between cattle and pig or mice – other mammals that STEC causes AE lesions, but I do understand that there are no such studies, yet.

Congrats again for your article – looking forward to see this online

Author Response

L45: to an unfamiliar reader this sounds like all cattle carrying EHEC shed the bacterium, however this is not the case as not all cattle carrying EHEC shed it. Maybe make it a bit more clear?

Response: The authors thank the reviewer for the positive comments on our study. To clarify that only some, but not all, cattle carrying STEC shed the bacterium, we added the following sentence. See L45: “Some cattle colonized with STEC shed this bacterium in feces, but shedding frequencies and concentrations can greatly vary across herds.”

L50-51: add a reference to this statement

Response: The authors thank the reviewer for noting that this statement should include a reference. We have modified this sentence and added a reference. See L51: “Though it is well known that intimin and the translocated intimin receptor are involved in adherence and attaching-effacing lesion formation [22], our knowledge of initial colonization factors is limited.”

  1. Vlisidou, I.; Dziva, F.; La Ragione, R.M.; Best, A.; Garmendia, J.; Hawes, P.; Monaghan, P.; Cawthraw, S.A.; Frankel, G.; Woodward, M.J. Role of intimin-tir interactions and the tir-cytoskeleton coupling protein in the colonization of calves and lambs by Escherichia coli O157: H7. Infect. Immun. 2006, 74, 758-764.

L66: maybe add a few details about the different days the tissues were obtained from the steers

Response: As suggested this sentence was modified to add the specific days the tissues were collected. See L69: “In 2017, tissues were obtained from three 18 month old steers of approximately 1,200 lbs on three separate days (June 14, June 30, and July 24) with each animal representing a single experiment.”

L87: I am a bit curious on why you incubated for 2h (and not more or less). This is probably something I do not know of but 1-2 more sentences would help non-specialists reading the article. Is it 2h because 1h is not enough and more than 2h is not necessary?

Response: The manuscript was revised to address the selection of a 2 h incubation time. See L122: “The proposed timeline of STEC adherence begins with initial interaction with the extracellular membrane [30], which was the focus of our study. To quantify this early interaction, a total of three bovine colonic explants were inoculated with E. coli strains and bacterial adherence was evaluated after 2 h. Previous studies demonstrated that 2 h is sufficient to produce initial adherence factors such as fimbriae and pili in E. coli and to allow for attachment to cell cultures [30,31].”

L104: raw? Reads were trimmed

Response: The reviewer is correct that these were raw reads. This sentence has been modified to include the phrase “raw reads”. See L109: “Raw reads were trimmed for adapters and quality using trim_galore version 0.4.5.”

L138-139: related to my previous comment: is the fact that 3h incubation shows similar results suggesting that 2h is enough for the initial STEC colonization? (question/just thinking out loud here): are there any studies that used a time series experiment to show how the adherence progresses

Response: To our knowledge there are no studies that have systematically evaluated colonization using a time series experiment. Longer incubation times such as 6 to 9 h have been evaluated (Baehler and Moxley, FEMS Microbiol Lett 2000 and 2002), but primarily for determining attaching-effacing lesion formation and explant tissue integrity.

L169-173: personally, I find the biotic/abiotic comparison a bit far fetched – but please do not remove it on my account; I simply say that I would be more interested in a comparison between cattle and pig or mice – other mammals that STEC causes AE lesions, but I do understand that there are no such studies, yet.

Response: The authors appreciate the reviewer’s comment and do not want to detract from the focus of the work being on colonization of the bovine colon. However, we have left the statement on the potential role that flagellum may play in attachment to spinach and glass as flagellum may be a common factor to colonize different environments.

Reviewer 2 Report

The authors have analysed the transcriptomes of 3 STEC strains adherent to bovine colonic explants and compared them to that of an E. coli K12 strain. They previously quantified the adhesion of these strains to the bovine colonic explants. This is an original peace of work as, to my knowledge, it is the first quantification of the adhesion of STEC and/or EHEC strains to bovine gut explants, and the first transcriptome analysis under these conditions. The work seems to have been conducted very carefully and the methods used are adapted to the objective. The results have been well analysed.

Unfortunately, the results are disappointing as only one gene is found significantly upregulated between EDL933 and K12. This is probably due to high variability between the samples (explants). The authors should have used more biological replicates to get better reliability and statistical significance.

Specific comments

Materials and methods section:

Why having used 37°C for incubation of E. coli strains with explants. I would have used 39°C which is considered the bovine internal temperature. However, this is just a comment, the authors do not have to conduct the experiments again.

I think that some details must be added to this section (see below).

Could you please indicate the origin of DEC10E (O26:H11, stx1, eae) and DEC8B (O111:H8, stx1, stx2, eae) strains (isolated from bovine feces?, from human clinical cases?...)

What was the age (or weight) of the steers from which the colonic explants were obtained?

Can you precise which part of the colon has been used for the explants

The authors should explain why they chose an incubation time of 2h. Have you conducted preliminary kinetics experiments to determine the time necessary for significant adhesion of the E. coli strains?

Results and discussion:

The Deseq-2 analysis found very few genes as significantly up-(or down) regulated. The authors indicate that variability between explants may explain this result. However, other factors may also contribute to such small differences in gene expression between the strains: incubation in a rich medium (RPMI 1640 complete medium), which does not represent the composition of the colon content, or the time used for incubation (2h).

Author Response

The authors have analysed the transcriptomes of 3 STEC strains adherent to bovine colonic explants and compared them to that of an E. coli K12 strain. They previously quantified the adhesion of these strains to the bovine colonic explants. This is an original peace of work as, to my knowledge, it is the first quantification of the adhesion of STEC and/or EHEC strains to bovine gut explants, and the first transcriptome analysis under these conditions. The work seems to have been conducted very carefully and the methods used are adapted to the objective. The results have been well analysed.

Unfortunately, the results are disappointing as only one gene is found significantly upregulated between EDL933 and K12. This is probably due to high variability between the samples (explants). The authors should have used more biological replicates to get better reliability and statistical significance.

Response: The authors agree with the reviewer that more explant samples may have reduced the variability and let to more significant differences between strains. We used 3 explant samples as is common in the literature for bovine explant studies and because of the limited availability of access to tissues. As the reviewer suggested below, other sources may have accounted for variability. Thus, we revised the manuscript to address various sources. See L231: “In addition, the use of the complex RPMI 1640 medium during incubation may have contributed to the homogenous response observed by RNA-seq. Also, evaluation of only one incubation time may have resulted in similar responses between strains. To our knowledge, no study has systemically evaluated a time-series experiment to determine STEC adherence and gene expression over time during contact with bovine explants, which warrants future investigation.”

Specific comments

Materials and methods section:

Why having used 37°C for incubation of E. coli strains with explants. I would have used 39°C which is considered the bovine internal temperature. However, this is just a comment, the authors do not have to conduct the experiments again.

Response: We used 37°C because it has been validated as a temperature to culture and infect bovine explants (Reference #13. Baehler, A.A.; Moxley, R.A. Escherichia coli O157:H7 induces attaching-effacing lesions in large intestinal mucosal explants from adult cattle. FEMS Microbiol. Lett. 2000, 185, 239-242.). We agree with the reviewer that 39°C would be more physiological relevant to the bovine internal temperature. We have added this as a limitation of our study. See L146: “The bovine internal temperature is 39°C, however explants were cultured and infected at 37°C based on established conditions in a previous study [13], which may be a limitation of our study in mimicking physiological conditions.”

I think that some details must be added to this section (see below).

Could you please indicate the origin of DEC10E (O26:H11, stx1, eae) and DEC8B (O111:H8, stx1, stx2, eae) strains (isolated from bovine feces?, from human clinical cases?...)

Response: The origin of each STEC strain has now been included. See L63: “The STEC strains DEC10E (O26:H11, stx1, eae) isolated from a calf [23], DEC8B (O111:H8, stx1, stx2, eae) isolated from a human patient [23], and EDL933 (O157:H7, stx1, stx2, eae) isolated from ground beef [24] were obtained from Dr. Shannon Manning (Michigan State University STEC Center).”

What was the age (or weight) of the steers from which the colonic explants were obtained?

Response: As the reviewer requested, the age and approximate weight of steers is now included. See L69: “In 2017, tissues were obtained from three 18 month old steers of approximately 1,200 lbs on three separate days (June 14, June 30, and July 24) with each animal representing a single experiment.”

Can you precise which part of the colon has been used for the explants

Response: The precise part of the colon that was removed was added to the manuscript. See L71: “A 0.3 m section of the centripetal coil was taken near the junction of the centrifugal and centripetal coils of the spiral colon.”

The authors should explain why they chose an incubation time of 2h. Have you conducted preliminary kinetics experiments to determine the time necessary for significant adhesion of the E. coli strains?

Response: Most studies with bovine explants have evaluated attaching-effacing lesions, which requires a 6 h incubation. For our study, we wanted to evaluate initial adherence which occurs prior to tight adherence and lesion formation. Thus, the selection of 2 h was based on previous studies that have determined that 2 h is sufficient to produce initial adherence factors such as fimbrae and pili and for attachment to cell cultures. To clarify this point in the manuscript, we added a section in the results. See L122: “The proposed timeline of STEC adherence begins with initial interaction with the extracellular membrane [30], which was the focus of our study. To quantify this early interaction, a total of three bovine colonic explants were inoculated with E. coli strains and bacterial adherence was evaluated after 2 h. Previous studies demonstrated that 2 h is sufficient to produce initial adherence factors such as fimbriae and pili in E. coli and to allow for attachment to cell cultures [30,31].”

Results and discussion:

The Deseq-2 analysis found very few genes as significantly up-(or down) regulated. The authors indicate that variability between explants may explain this result. However, other factors may also contribute to such small differences in gene expression between the strains: incubation in a rich medium (RPMI 1640 complete medium), which does not represent the composition of the colon content, or the time used for incubation (2h).

Response: We agree with the reviewer that other factors may have contributed to the small differences in gene expression between strains. We have modified the manuscript to include medium composition and incubation time. See L231: “In addition, the use of the complex RPMI 1640 medium during incubation may have contributed to the homogenous response observed by RNA-seq. Also, evaluation of only one incubation time may have resulted in similar responses between strains. To our knowledge, no study has systemically evaluated a time-series experiment to determine STEC adherence and gene expression over time during contact with bovine explants, which warrants future investigation.”